# Evapotranspiration in Semi-Arid Climate: Remote Sensing vs. Soil Water Simulation

**DOI:** 10.3390/s23052823

**Published:** 2023-03-04

**Authors:** Hedia Chakroun, Nessrine Zemni, Ali Benhmid, Vetiya Dellaly, Fairouz Slama, Fethi Bouksila, Ronny Berndtsson

**Affiliations:** 1LR99ES19 Laboratory of Modelling in Hydraulics and Environment (LMHE), National Engineering School of Tunis (ENIT), University of Tunis El Manar, BP 37, Tunis 1002, Tunisia; 2LR20INRGREF04 Laboratory of Rural Engineering, National Institute for Research in Rural Engineering, Water and Forestry (INRGREF), University of Carthage, BP 10, Ariana 2080, Tunisia; 3Division of Water Resources Engineering, Lund University, P.O. Box 118, SE-221 00 Lund, Sweden; 4Centre for Advanced Middle Eastern Studies, Lund University, P.O. Box 201, SE-221 00 Lund, Sweden

**Keywords:** S-SEBI, HYDRUS-1D, 5TE sensor, evaporative fraction, evapotranspiration, barley, potato

## Abstract

Estimating crop evapotranspiration (ET_a_) is an important requirement for a rational assessment and management of water resources. The various remote sensing products allow the determination of crops’ biophysical variables integrated in the evaluation of ET_a_ by using surface energy balance (SEB) models. This study compares ET_a_ estimated by the simplified surface energy balance index (S-SEBI) using Landsat 8 optical and thermal infra-red spectral bands and transit model HYDRUS-1D. In semi-arid Tunisia, real time measurements of soil water content (θ) and pore electrical conductivity (EC_p_) were made in the crop root zone using capacitive sensors (5TE) for rainfed and drip irrigated crops (barley and potato). Results show that HYDRUS model is a fast and cost-effective assessment tool for water flow and salt movement in the crop root layer. ET_a_ estimated by S-SEBI varies according to the available energy resulting from the difference between the net radiation and soil flux G_0_, and more specifically according to the assessed G_0_ from remote sensing. Compared to HYDRUS, the ET_a_ from S-SEBI was estimated to have an R^2^ of 0.86 and 0.70 for barley and potato, respectively. The S-SEBI performed better for rainfed barley (RMSE between 0.35 and 0.46 mm·d^−1^) than for drip irrigated potato (RMSE between 1.5 and 1.9 mm·d^−1^).

## 1. Introduction

Crop productivity must be increased through modern agricultural processes, while other natural resources and the environment need to be protected to meet rapidly expanding food demands [1]. Precision agriculture is an innovative management concept that is becoming increasingly important in arid and semi-arid areas due to climate change and increasing water salinity risks. It aims to align land management with crop needs to improve production, reduce environmental damage, and raise agricultural product quality standards using innovative methodologies and technologies for data acquisition and processing [2]. For example, smart irrigation is a system of automatically started/interrupted irrigation associated with using dielectric soil sensors. Dielectric sensors are often used in agriculture for real-time soil properties monitoring. Time domain reflectometry (TDR) has been established as the most precise dielectric technique to measure volumetric water content (θ) and bulk electrical conductivity (ECa) in soils providing automatic, simultaneous, and continuous reading. Topp et al. [3] reported its first application to soil water measurements. The efficiency of TDR in agriculture has led to the development of other dielectric techniques, such as the frequency domain reflectometry (FDR) method, which is nowadays considered as an accurate low-cost technique [4,5].

One of the most critical factors in precision agriculture is crop water requirements based on actual evapotranspiration ET_a_. Estimation of ET_a_ is performed by either empirical methods with the most widely used FAO-56 crop coefficient method [6] or by residual methods based on the surface energy balance (SEB), which controls water exchange and partitioning into sensible and latent heat in the soil-vegetation-atmosphere continuum [7]. The SEB models combine remote sensing of surface radiometric temperature, albedo, and vegetation cover with meteorological data to determine the latent and sensible heat fluxes. SEB models differ in data requirements, and the way of considering the complex soil and vegetation system, by either using single-source models when soil and vegetation fluxes are not separated, or dual-source models that are considered more appropriate for partially vegetated areas [8]. SEB models can also be classified into contextual models considering the contrast between wet and dry pixels, or pixel-based models [9]. Amongst the single-source contextual models, S-SEBI [10] determines the evaporative fraction Λ representing the ratio between the latent heat flux and the sum of sensible and latent heat flux. S-SEBI is the simplest SEB algorithm since Λ is determined empirically from the image, and it does not require meteorological input data. It is particularly adapted to semi-arid and heterogeneous fields, as described by [10,11,12].

The estimated fluxes by SEB models are commonly based on in situ measurements of the eddy covariance or lysimeter variables, which may be expensive and require frequent maintenance. Physical models estimating soil water balance (SWB) used in combination with SEB methods are either based on steady or transient states. In the first case, the FAO-56 crop coefficient method has been widely used to estimate vegetation transpiration and soil evaporation from remotely sensed vegetation indices and soil characteristics. The FAO-56 method has been the basis of many operational applications for irrigation monitoring [13,14,15]. Transient models often require soil hydraulic properties inputs to calibrate mechanistic processes either to determine the transpiration [16] or the evapotranspiration [17,18]. On the other hand, less attention, however, has been paid to the estimation of ET_a_ by SEB models compared to transient methods, such as the HYDRUS model. HYDRUS-1D is considered an accurate model for soil water flow simulation by including different components of the plant-soil hydrological process [4,19,20,21]. 

In view of the above, the objective of this study was to assess the performance of the S-SEBI model using remotely sensed data to determine ET_a_ for irrigated potato and rainfed barley, in comparison to modeling the unsaturated water flow using HYDRUS-1D. The two different approaches were compared to observed soil water content. Improved estimation of ET_a_ is increasingly crucial due to increasing water scarcity in arid and semi-arid areas. Most of these agricultural systems are rainfed but some important crops are irrigated. Thus, we covered both rainfed and irrigated systems in our study. The experimental area was an agricultural region located in northeastern Tunisia.

## 2. Materials and Methods

### 2.1. Study Area and Data Collection

The experimental study area is in the Cape Bon region in northeastern Tunisia. It is an agricultural area with a semi-arid climate. The mean annual precipitation and potential evapotranspiration are 450 mm and 1370 mm, respectively (1982–2006) [4]. The driest and wettest months are July and January, respectively. The experimental Oued Souhil plot belongs to the National Institute of Water, Forest, and Rural Engineering (INRGREF) in Nabeul (36°28′43.16″ N, 10°42′24.84″ E). The area around the experimental plot comprises urban, agriculture, forest, and bare soil fields with rainfed and irrigated water supply (Figure 1). Soil properties were determined from samples collected at every 20 cm depth to 100 cm measuring soil water content, soil electrical conductivity, bulk density, and pH according to USDA methods [22]. In the study area, the soil is mainly sandy loam (Table 1), and the water table is located at about 9 m below the soil surface. 

#### 2.1.1. Crop Water Need

The experimental plot was approximately 500 m^2^ and the selected crops during the investigation period (2015–2017) were irrigated potato (2015–2016) and rainfed barley (2017). The irrigation water salinity EC_w_ was around 3.5 dSm^−1^. A surface drip irrigation system was used for potato during two cropping seasons (2015 and 2016) while no irrigation was applied for barley. The CROPWAT 8.0 software was used to estimate the crop water need. The software requires climatic, soil, and crop-specific coefficient K_c_ data to estimate the irrigation need. For this purpose, the Penman–Monteith equation [23] was used to estimate the reference evapotranspiration rate, ET_0_ (mm), based on daily radiation, air temperature, air humidity, and wind speed, collected from an agro-climatic station at the experimental plot. The crop specific coefficients were derived from [6] (Table 2).

During the first late season potato growing period that lasted 101 days (from 11 March 2015 to 7 July 2015), the amount of irrigation was about 336 mm allocated in 16 applications together with 20 mm of rainfall. The second early season potato growing period lasted 120 days (from 18 January 2016 to 17 May 2016) with 170 mm irrigation allocated in 8 applications and rainfall of roughly 70 mm. During the rainfed barley cropping season, which lasted from 18 October 2016 to 20 June 2017, the total precipitation was approximately 291 mm (Table 2).

The FAO-56 method [6,24] uses the reference evapotranspiration (ET_0_) that depends on atmospheric demand to estimate crop water requirements (CWR), approximated by the Penman–Monteith equation. When there is no water limitation, the crop water requirement equals the actual crop evapotranspiration, considering the crop characteristics and the soil factors. The FAO-56 can be expressed in two alternatives: the single or dual crop coefficient. In the dual crop coefficient variant, two coefficients are used: one related to the vegetation transpiration K_cb_ varying throughout its growing season, and the other to the soil evaporation K_e_. When considering the reduction of crops transpiration due to either water shortage available for roots or water salinity, the basal crop coefficient K_cb_ is multiplied by a stress coefficient K_s_ as expressed in Equation (1). Under unrestricted water availability and quality, one can use the single crop coefficient to determine ET_c_ based on a unique crop coefficient K_c_ combining both vegetation transpiration and soil evaporation (Equation (2)).
(1)$${ET}_{a}=\left(K_{s}K_{cb}+K_{e}\right){ET}_{0}$$
(2)$${ET}_{c}={K_{c}ET}_{0}$$

#### 2.1.2. Soil Water Content Monitoring 

Capacitive sensors (5TE-Decagon Devices Inc., Pullman, WA, USA) were used for hourly real time monitoring of soil water content (θ), pore soil water electrical conductivity EC_p_, and soil temperature (T_soil_). The Decagon Em50 data logger and DATATRAC software were used for data saving and downloading, respectively (Figure 2). The 5TE probe is a frequency domain reflectometry sensor, oscillating at 70 MHz. The sensor measures soil apparent permittivity (K_a_) soil bulk electrical conductivity (EC_b_), and soil temperature T_soil_ [5]. Models were used for θ and EC_p_ estimation, [3,25]. Four 5TE sensors were installed in the root zone at four different soil depths (0.05 m, 0.25 m, 0.45 m, and 0.90 m) for both crops (Figure 2). The investigated soil depths were chosen to reflect the shallow root system of both potato and barley crops [6].

#### 2.1.3. Earth Observation Data

A total of 26 cloud free Landsat 8 Operational Land Imager (OLI) and thermal infrared sensor (TIRS) images between 1 June 2015 and 23 August 2017 were acquired (http://earthexplorer.usgs.gov/ accessed on 3 March 2023). Landsat 8 images require preprocessing to determine the top of canopy (TOC) reflectance used in the calculation of biophysical variables free from atmospheric effects. The MODTRAN (moderate resolution atmospheric transmission) algorithm based on radiative transfer [26] was used to produce the reflectance bands of the Landsat 8 images. The albedo of the Earth’s atmosphere is the fraction of solar energy that is reflected to space. It is calculated from the TOC reflectance of the images spectral bands (B2 to B7) by Equations (3) and (4). The processed images were used to assess the albedo and the NDVI (Normalized Difference Vegetation Index) (Tucker, 1979) calculated from red (R) and near infra-red (NIR) spectral bands (Equation (5)). Thermal bands B10 and B11 were corrected from atmospheric effects to obtain the surface temperature (T_s_) at 100 m resolution.
(3)$$\alpha =\sum \left(w_{b}\rho \left(b\right)\right)$$
(4)$$\rho \left(b\right)=\frac{\pi L\left(\lambda \right)d^{2}}{K^{↓}\left(b\right)cos\left(\vartheta_{z}\right)}$$
(5)$$NDVI=\frac{\rho_{NIR}-\rho_{R}}{\rho_{NIR}+\rho_{R}}$$
where

w_b_: weight bands B2 to B7 

ρ(b): reflectance in spectral band b

L(λ): spectral radiance for wavelength λ [W·m^−2^·sr^−1^·μm^−1^]

d: distance between the Sun and the Earth [m]

K^↓^ (b): incoming short-wave radiation at top of atmosphere (TOA) for band b

θ_z_: solar zenith angle [rad]

### 2.2. ET_a_ Estimation Approaches

#### 2.2.1. ET_a_ by S-SEBI

The SEBI method (surface energy balance index) was proposed by Menenti and Choudhury [27] to determine the evapotranspiration by solving the surface energy balance considering the contrast between dry and wet limits determined partially or completely from remote sensing data. The SEBI model uses the energy balance (Equation (6)), where the total available energy (R_n_) coming from the Sun and the atmosphere is transformed into soil heat flux (G_0_), sensible heat flux (H) relative to the surface transfer to the environment, and latent heat flux (λE), where λ is the energy required to change a unit mass of water from liquid to water vapor in a constant pressure and constant temperature process. The net radiation (R_n_) is determined from remote sensing variables acquired in the visible and infra-red spectral bands for albedo calculation, and the thermal infrared for surface temperature calculation (Equation (7)). The surface emissivity (ε_s_) is considered either constant over the whole study region and in time ([28], ε_s_ = 0.98) or as a function of vegetation indices [29].
(6)$$R_{n}=G_{0}+H+\lambda E$$
(7)$$R_{n}=\left(1-\alpha \right)R_{g}-\sigma \varepsilon_{s}T_{s}^{4}+L^{↓}$$
where

R_n_: net radiation at the crop surface [W·m^−2^]

α: surface albedo [−]

R_g_: solar radiation [W·m^−2^]

ε_s_: surface emissivity [−] 

σ: Stefan–Boltzman constant [−]

T_s_: surface temperature [°K]

L^↓^: long wave incoming radiation [W·m^−2^] (calculated by $\sigma \varepsilon_{a}T_{a}^{4}$, where ε_a_ is the atmosphere emissivity, and T_a_ is the air temperature)

Sugita and Brutsaert [30] assumed the evaporative fraction Λ representing the ratio between the latent heat flux and the summation of sensible and latent heat flux (Equation (8)), to be constant during the daylight hours to determine regional daily ET_a_. This fraction is an indicator of the hydrological moisture situation.
(8)$$\Lambda =\frac{\lambda E}{\lambda E+H}=\frac{\lambda E}{R_{n}-G_{0}}$$

The S-SEBI algorithm, a simplified version of SEBI, is based on dry and wet limits to partition available energy into sensible and latent heat fluxes [9,10]. In constant atmospheric conditions, the temperature and reflectance are correlated, and it is then possible to distinguish two states (Figure 3):-For saturated soil, such as irrigated areas, the temperature is almost constant at low T_s_ because available energy is used for evaporation. At higher reflectance, T_s_ increases because of decreasing ET_a_ due to less soil moisture. This is the “evaporation controlled” state corresponding to the dry edge of the scatter-At an inflexion threshold reflectance, the surface temperature decreases with increasing reflectance because no more evaporation can take place. This is the “radiation controlled” state corresponding to the wet edge of the scatter.

In the S-SEBI algorithm, for a given pixel with (α,T_s_), the evaporative fraction Λ is based on the maximum and minimum surface temperatures (T_H_ and T_λE_) corresponding to the dry edge with a maximum sensible heat flux (H_max_), and to the wet edge with a maximum latent flux (λE_max_), respectively, as illustrated in Figure 3. The extreme temperatures are determined from the scatter of the image, thus allowing a simplified version of the SEBI method. The Λ is therefore given in the S-SEBI algorithm shown in Equation (9).
(9)$$\Lambda =\frac{a_{H}+b_{H}(\alpha )-T_{s}}{a_{H}-a_{\lambda E}+\left(b_{H}-b_{\lambda E}\right)(\alpha )}$$

The soil flux G_0_ is generally considered as a residual term of the energy balance equation or assumed to be negligible on daily timescales [8]. It is commonly parameterized as a constant (R_n_; referred to as the ratio G_0_/R_n_). However, it varies with vegetation cover, and can range from 0.10 for full cover [31] to a value between 0.10 and 0.50 over sparse canopies and bare soils [32,33]. Soil properties relative to water retention also affect the distribution of the available energy between both sensible and soil heat fluxes [33,34]. In SEB algorithms, various relations are proposed to determine G_0_/R_n_ based on surface temperature, albedo, reflectance in near infra-red and in red, and vegetation indices, such as the NDVI or the modified soil adjusted vegetation index MSAVI [35]. In Equations (10)–(13) different formulas of G_0_ calculation are given as a proportion of R_n_ as proposed by Clothier [36], Daughtry et al. [37], Bastianssen et al. [38], and Sobrino et al. [39].
(10)$$G_{0}=(0.295-0.01331\frac{\rho_{NIR}}{\rho_{R}})R_{n}$$
(11)$$G_{0}={\left(\frac{T_{s}-273.15}{\alpha }\right)\left(0.032\alpha +0.062\alpha^{2}\right)\left(1-0.978{NDVI}^{2}\right)R}_{n}$$
(12)$$G_{0}=\left(T_{s}-273.15\right)\left(0.0038+0.074\alpha \right)\left(1-0.98{NDVI}^{4}\right)R_{n}$$
(13)$$G_{0}=0.5Exp(-2.13MSAVI)R_{n}$$

The instantaneous latent heat flux λE is converted to ET_i_ (mm·h^−1^) considering the amount of energy to evaporate a unit weight of water (2,454,000 J·kg^−1^). The scaling of the ET_i_ derived from remotely sensed surface temperature and reflectance at a single time of the day to daily ET_a_ has been discussed in [9]. The model proposed by [40] relating instantaneous to daily ET_a_ in mm·d^−1^ assumes that the diurnal trend of λE follows the course of solar radiation during day light as given by [41] (Equation (14)).
(14)$${ET}_{a}={ET}_{i}\left(\frac{2N}{\pi \sin\frac{\pi t}{N}}\right)$$
where

N: the total time from sunrise to sunset [h]

t: the time elapsed since sunrise [h].

#### 2.2.2. ET_a_ by HYDRUS-1D Transient State Model

##### Modelling Approach

Mechanistic approaches have been widely used to simulate the soil water balance components [4,21,42,43]. Nowadays, the HYDRUS-1D model is generally recognized as an effective and precise conceptual model for simulating water flow in variably saturated soils under different irrigation methods. The model is based on the Richards equation (Equation (15)) and root-soil interaction processes. The unsaturated soil hydraulic properties are described with a set of closed-form equations resembling those of van Genuchten [44] (Equations (16) and (18)) who used the statistical pore-size distribution model of Mualem [45] (Equation (17)) to obtain a predictive equation for the unsaturated hydraulic conductivity function.
(15)$$\frac{\partial \theta }{\partial t}=\frac{\partial }{\partial z}\left[K\left(\frac{\partial h}{\partial z}+1\right)\right]-S$$
where

h: water pressure head [m] 

θ: volumetric water content [m^3^·m^−3^] 

t: time [day]

z: spatial coordinate [m] (positive upward) 

S: sink term [m^3^·m^−3^·d^−1^] (root water uptake) 

K: unsaturated hydraulic conductivity function [m·d^−1^]
(16)$$\theta \left(h\right)=\left\{\begin{array}{c} \theta_{r}+\frac{\theta_{s}-\theta_{r}}{{\left(1+{\left|\alpha h\right|}^{n}\right)}^{m}}\;h<0\\ \theta_{s}\;\;\;\;\;h\geq 0\; \end{array}\right.\;\;$$
(17)$$K(h)=K_{S}{S_{e}}^{0.5}{\left[1-{\left(1-{S_{e}}^{1/m}\right)}^{m}\right]}^{2}$$
(18)$$Se=\frac{\theta -\theta_{r}}{\theta_{s}-\theta_{r}}$$
where 

θ_s_: saturated water content [m^3^·m^−3^]

α, n, m (= 1 − 1/n, n > 1) are empirical parameters [m^−1^], [−], [−] 

θ_r_: residual water content [m^3^·m^−3^] 

Se: effective water content [−] 

K_S_: saturated hydraulic conductivity [m·d^−1^]

Root water uptake is a sink term in the Richards equation (Equation (15)) defined as the volume of water removed by the plant from a unit volume of soil per unit time. The root water uptake rate is calculated from crop transpiration rate associated with rooting depth and soil water potential [46]. In the present study, HYDRUS-1D was used to estimate the actual daily crop evapotranspiration for potato and barley crops based on the soil water balance (SWB). The water flow and transport parameters used were calibrated and validated by [4] working on the same experimental plot of Oued Souhil during potato cropping season (2015) (Table 3). Given that the model has been calibrated and validated on the basis of hourly observed measurements of soil water content and salinity, it is considered as a reference for comparison among the different scenarios.

##### Boundary, and Initial Conditions 

The 1D model geometry was defined as a 1 m deep soil subdivided into five 0.2 m layers. Atmospheric boundary condition with surface runoff constituted the upper boundary. The HYDRUS-1D atmospheric boundary condition requires inputting separately evaporation and transpiration. To split evapotranspiration ET_a_ to actual evaporation (E) and actual transpiration (T), we applied Equations (19)–(21) proposed by Ritchie [47] where evapotranspiration, evaporation, and transpiration are expressed in mm.
(19)$$T={ET}_{c}.\mathrm{SCF}$$
(20)$$E={ET}_{c}(1-SCF)$$
(21)$$\mathrm{SCF}=1-e^{-0.46LAI}$$

The surface cover fraction (SCF) was estimated from the leaf area index (LAI), a critical biophysical variable affecting land surface processes, such as photosynthesis, transpiration, and energy balance. For the potato crop, the LAI values were derived from the literature and introduced according to the observed stages of potato cycle. LAI values ranged between 1 and 3.5 m^2^m^−2^ [48] (scenario-HP1). For the barley crop, two scenarios were studied, the LAI values were first derived from the literature and introduced according to the observed stages of barley cycle [49] (senario-H1B), and secondly LAI was estimated from observed NDVI-LAI relationship (scenario-H2B). This relationship is based on [50] where measured LAI with hemispherical photography were compared to Sentinel 2 NDVI (Equation (22)). The maximum and minimum LAI values for barley were equal to 3.5; 0.29 m^2^m^−2^ and 3.1; 0.092 m^2^.m^−2^ for H1B and H2B scenarios, respectively.
(22)$$LAI=e^{\left(-4.58+7.33NDVI\right)}$$

As groundwater in the study area is deep (about 9 m), we did not expect significant capillary rise from deeper layers in our profile and therefore, we set the lower boundary condition as free drainage. For solute upper boundary condition, a third-type Cauchy boundary condition was considered, and concentration flux was attributed. Salinity of observed irrigation water was, therefore, assigned at the top and zero concentration gradient for solute bottom boundary. Water content as well as soil salinity initial conditions were derived from observed data.

##### Root Water Uptake and Root Depth

Different mathematical models have been developed to simulate the process of water uptake by plant roots. These models typically take into account the physical properties of soil, such as hydraulic conductivity and water retention, as well as the plant’s physiological characteristics, such as root distribution and water transport properties. Accurate modeling of water uptake by plant roots can provide insights into plant growth and productivity, and can also help inform management strategies for agricultural and forestry systems [51].

In the Richards equation (Equation (15)), the sink term (S) represents the water uptake by crop roots and is defined as the volume of water removed from a unit volume of soil per unit time due to plant water uptake (Equation (23)) [46].
(23)$$S\left(h\right)=\alpha \left(h\right)Sp$$
where the root-water uptake water stress response function (α(h)) is a prescribed dimensionless function of the soil water pressure head (0 ≤ α ≤ 1), and Sp is the potential water uptake rate [s^−1^]. According to [46], water uptake is assumed to be zero close to saturation and for h lower than the wilting point pressure head. Water uptake decreases (or increases) linearly with h. For both crops, the Feddes model was considered as the water stress response function for root water uptake, and parameters for potato and barely crop, available in HYDRUS-1D, were selected [46].

Root depth was set equal to 60 cm and 100 cm for potato and barley, respectively, as defined by the FAO [24]. The adopted irrigation scheduling, use of brackish irrigation water, and initial soil salinity could affect the potential root water uptake due to both osmotic and water stress effects. Therefore, these stresses were considered to be multiplicative for both crops. The osmotic effects of pore water salinity on root water uptake were considered by applying a threshold-slope function suggested in the HYDRUS-1D crop parameters database with a slope of 6% and a threshold value EC_t_ of 3.40 dSm^−1^ for potato and a slope of 2.5% and a EC_t_ of 16 dSm^−1^ for barley [52]. In Table 4 we provide a description of datasets, sources, and the preprocessing stages required for the inputs to the different models studied. For each crop, the results of various scenarios of ET_a_ computation are evaluated in the next section.

## 3. Results and Discussion

### 3.1. Energetic Fluxes Assessment by S-SEBI 

The dry and wet edges were derived from the scatter in the space (α, T_s_) corresponding to crop pixels within the agricultural area around the experimental station, which is a square area of 12 × 12 km. An NDVI mask was determined for each Landsat 8 image date to avoid considering non-vegetation pixels (Figure 4). The scatter of vegetated crop areas is close to the trapezoidal shape in most development crop dates. This scatter comprises both forested and vegetated pixels, however we can easily separate them as the forest pixels correspond to the left part of the scatter (low albedos). The performance of S-SEBI model depends on the existence of various water status of pixels within the used image [10], as shown by NDVI classes in Figure 4. The manual method was used to define lines delimiting the extreme edges to which each pixel is being evaluated to determine its evaporative fraction. These edges correspond to the right side of the trapeze scatter comprising agriculture pixels. The dry edge was defined by pixels localized on the steep side of the trapeze reflecting the difference in T_s_ between dry and wet pixels at a given date, whereas the wet edge corresponds to the low base of the trapeze generally characterized by a gentle slope. The resulting evaporative fraction Λ is shown in Figure 5 as well as the variation of NDVI derived from time series images Landsat 8 (30 m resolution) and NDVI derived from Sentinel 2 (10 m resolution) in the experimental plot. The vegetation indices reflect the crop dynamic during the crop growing season based on the reflectance of photo-synthetically active plant covers in the visible and the infra-red spectral ranges. The spectral responses of vegetation for these wavelengths, and the resulting NDVI and LAI, are suited to specific growth conditions at the plot scale for monitoring crops throughout their development cycle, as reported in studies showing the existence of linear relations between crop coefficient and theses spectral response [53,54,55,56].

The full crop development corresponding to NDVI larger than 0.5 is characterized by high Λ values in the case of water availability, whereas hydric stress is highlighted by low energy used for evapotranspiration λE compared to the available energy (R_n_-G_0_). For both early and late potato growing seasons, the evaporative fraction does not exceed 0.6, while for rainfed barley it reaches 0.8 (Figure 5).

The calculation of soil heat G_0_ was based on the proportionality between G_0_ and R_n_ through a coefficient determined by the surface vegetation and soil properties (Equations (10)–(13)). Compared to measured soil moisture at image acquisition dates, the range of the G_0_/R_n_ ratio variation illustrated in Figure 6 shows that G_0_ increases as the soil dries, thus leaving more energy to the sensible and latent flux since the available energy (R_n_-G_0_) decreases. Based on a model simulating heat and water movement through the canopy, it was shown by [33] that differences in available energy can reach 200 W.m^−2^ between the various parametrization of soil conditions (texture and wetness). Compared to the most used ratio in the literature (G_0_/R_n_ = 0.2), the four models tested in our study show a G_0_–RMSE between 27 and 113 W.m^−2^ for rainfed barley and between 25 and 117 W.m^−2^ for irrigated potatoes (Table 5). Santanello and Friedl [33] showed that for LAI varying from 0.1 to 5 m^2^.m^−2^ (sparse to full vegetation) in the case of sand-loam soil, simulations of the ratio G_0_/R_n_ vary between 0.2 and 0.4, thus leaving more available energy to heat the soil surface and increase both sensible and soil heat fluxes. We discovered that the closest variation of the ratio G_0_/R_n_ to 0.2 by the empirical models proposed by Sobrino et al. [39] and Clothier [36] compared to Bastianssen et al. [38] for whom this ratio varied between 0.25 and 0.45. We can observe that G_0_ values calculated by this model continue to grow while the two other models tend to decrease at the end of crop growth season (Figure 6). The model that deviates most from the other three is the one proposed by Daughtry et al. [37], which gives the highest G_0_/R_n_ coefficients between 0.3 and over 1. Based on experiments in previous studies, we kept the three models where the G_0_/R_n_ ratio did not exceed 0.5 to assess the difference in the flux determined by S-SEBI. As shown in Table 5, the variation of G_0_ (compared to a ratio of G_0_/R_n,_ which was equal to 0.2) reaches 113 W.m^−2^ for barley and 117 W.m^−2^ for potato, generating a variation in the λE between 25 to 42 W.m^−2^. Regarding these substantial differences in flux calculation by S-SEBI when considering different methods for G_0_/R_n_ assessment, the different scenarios of the energetic approach are compared to the calibrated HYDRUS-1D model in the experimental plot. For each crop we considered 3 scenarios of the S-SEBI model using Clothier [36], Bastianssen et al. [38], and Sobrino et al. [39] for G_0_/R_n_ calculation. These scenarios are referred to as S1P1, S2P1, and S3P1 for the late growing season of potato, S1P2, S2P2, and S3P2 for the early growing season of potato, and S1B, S2B, and S3B for barley (refer to scenarios in Table 4). 

### 3.2. ET_a_ Results for Irrigated Potato

#### 3.2.1. ET_a_ Estimated by HYDRUS-1D

Both SWB and HYDRUS models were used to continuously evaluate ET_a_ from sowing to harvest of potato for early and late seasons. The HYDRUS evapotranspiration results (HP1 and HP2 scenarios) and the FAO-56 (FP1 and FP2 scenarios) are presented in Figure 7, as well as the amounts of precipitation and net irrigation water for the respective irrigation schedules. HYDRUS and FAO-56 approaches both displayed similar ET_a_ dynamics during the studied period, although they do differ in their amounts. The two approaches present equal amounts, during the potato’s initial stage, up to the first and second irrigation events for early (March 2015, ET_a_–HP1) and late (January 2016, ET_a_–HP2) potato-growing season. Thereafter, ET_a_ by HYDRUS decreased under both water and solute stress effect, especially for the early potato season (March 2015, ET_a_–HP1). In fact, the most part of ET_a_–HP1 growing season occurs under solute stress (from 26 April) showing an increase in root zone salinity above potato EC_t_ limit (3.4 dSm^−2^), as shown by Figure 8. The decrease in ET_a_ is mainly explained by the inadequate irrigation scheduling and doses combined with brackish irrigation water [4,21,57]. ET_a_–HP results indicate that, when both water and solute stress are considered, sowing late potato (ET_a_–HP1) yielded in better water consumption (85% ET_a_) with respect to early potato (ET_a_–HP2-67% ET_a_). This fact is related to the late growing period (winter and spring), resulting in less salt stress mainly during the initial and the mid-crop stages (Figure 8). Similarly, Hu et al. [58] noted that in a dry year, late planting dates significantly improved water use efficiency.

In addition, we notice that the ET_a_–FP scenario did not display any effect of water and solute stress, operating at optimum conditions based on the K_c_ value derived from [6], while the ET_a_–HP scenario was based on the actual condition considering the effect of both water and solute stress. As a matter of fact, Han et al. [59] studied ET_a_ estimation approaches, and they reported that using only the standard crop coefficient (K_c_) recommended by FAO-56 would reduce prediction accuracy. The use of dual crop coefficients (Equation (1)) is more realistic in the case of irrigated crops as the stress coefficient (K_s_) considers not only the water shortage but also salinity effects on root water uptake [60]. The HYDRUS model includes different components of the plant-soil water interaction, and its boundary conditions are flexible, which makes it possible to simulate the real water cycle in the field [61]. Furthermore, Ghazouani et al. [62], working on potato crop in central Tunisia, confirmed relevant results of the HYDRUS-2D model for the assessment of potato transpiration under salt-water stress. 

#### 3.2.2. ET_a_ Estimated by S-SEBI 

Results of daily ET_a_ derived by energetic fluxes with S-SEBI compared to daily HYDRUS ET_a_ (during the days corresponding to Landsat 8 image) revealed a relatively good agreement with R^2^ values between 0.6 and 0.7, depending on the scenario adopted for the G_0_/R_n_ ratio. The results showed that S-SEBI correctly reflects the dynamics of ET_a_ during both potato seasons. However, compared to the simulated HYDRUS, S-SEBI underestimated ET_a_ with an average RMSE between 1.5 and 1.9 mm·d^−1^, depending on the scenario adopted. The least RMSE was found when G_0_/R_n_ ratio is close to 0.2 (scenarios S1P1, S3P1, S1P2, S3P2). Compared to widely accepted RMSE values by ET community between 0.2 and 0.9 mm·d^−1^ [63], the RMSE is relatively high. However, when using S-SEBI, one finds RMSE up to 1.71 mm·d^−1^ between modeled and measured eddy covariance ET_a_ in the case of barley (S-SEBI based on Landsat 8 in [64]), or an RMSE of the order of 1.3 mm·d^−1^ compared to HYDRUS ET_a_ in the case of vineyard (S-SEBI using ASTER images in [18]). It should be noted that few studies have assessed the ET_a_ of potatoes by remote sensing integrated into SEB-type models, although some have assessed the water stress index developed by Jackson [65] based on infrared thermography of potato [62,66], while others study the detection of the combined effect of drought and salinity on potato by optical remote sensing [67]. In our study, the underestimation of ET_a_ is due to the relatively low Λ (0.60 for early potato and 0.46 for late potato), as shown by Figure 5. This can be explained by the fact that HYDRUS simulates ET_a_ within the soil patch comprising the plant with an area of about 500 m^2^, thus covering almost a half Landsat 30 m pixel, whereas the satellite-based values are derived within mixed pixels comprising vegetation and soil (Figure 2). Moreover, the 100 m resolution pixel of thermal bands used for Ts estimation within the experimental plot is also too coarse. To overcome this issue, the disaggregation methods could be used to refine the remotely sensed land surface temperature as reported in [64,68] where regression models allowed the adjustment between variables from 30 m resolution bands (NDVI and albedo) and Ts estimation by upscaling the thermal bands to 30 m resolution. Also the possibility of combining Landsat 8 and Sentinel 2 data could also be tested in further works, as the multi-sensor fusion data improved the robustness of the time-series ET_a_ estimation based either on FAO-56 Kc estimation [69], or the estimation of the evaporative fraction through a machine learning process combining Landsat 8 and Sentinel 2 multiresolution products [70].

### 3.3. ET_a_ Results for Rainfed Barley

#### 3.3.1. ET_a_ Estimated by HYDRUS-1D

Figure 9 presents the temporal evolution of barley evapotranspiration estimated with HYDRUS-1D in scenarios H1B and H2B during growing season (2016/2017). The experiment was conducted under only rainfed condition; indeed, solute effect is minimal and not significant. The total precipitation during the initial crop growing season was 291 mm with only 71 mm for the end of season. Depending on LAI values, two scenarios were studied with the first (H1B) using LAI from the literature as a reference to derive the LAI course according to observed stages of the barley cycle [49] (scenario H1B), and the second scenario (H2B) with LAI estimated from the observed NDVI-LAI relationship. A good agreement between the two scenarios during the studied period was noted, except for the late crop stage (May 2017) where LAI values decreased from 3 to 0.09 m^2^·m^−2^ (H1B) and from 3.4 to 2.6 m^2^·m^−2^ (H2B). We observed that the degree of variation in LAI values is significant, with minor impact on HYDRUS outputs. Bufon et al. [71] studied drip irrigated cotton using the HYDRUS model with different LAI values and reported that the increase in LAI affected the calculation of root water uptake (HYDRUS-transpiration rate). These differences in results may be associated with various factors, such as irrigated or rainfed cropping systems. Compared to the conventional FAO-56 method, both HYDRUS scenarios were highly correlated with FB scenario up to April 2017, showing a coefficient of determination R^2^ equal to 0.98. After this date, R^2^ highly decreased to less than 0.1. This fact is strictly linked to soil water availability. In fact, according to the FAO-K_c_ model assuming optimum conditions, barley water need is 388 mm (100% ET_a_) while during the experiment period, the total rainfall was about 291 mm (74% ET_a_). Particularly, a water deficit was noted from April 2017, with total precipitation reaching only 17 mm, and considered as not sufficient to cover the water need for the subsequent crop stages, thus, effectively resulting in water stress. The impact of water stress on barley evapotranspiration is conditioned by both the intensity of the water deficit and the stage at which it occurs [72].

#### 3.3.2. ET_a_ Estimated by S-SEBI

Results of daily ET_a_ derived by energy fluxes with S-SEBI compared to daily HYDRUS ET_a_ revealed a good agreement between them, with high R^2^ around 0.86 for all scenarios adopted in relation to G_0_/R_n_ ratio. This shows that S-SEBI correctly reflects the dynamics of ET_a_ during the barley growing season. Compared to the simulated HYDRUS by the two LAIs and the three scenarios with varying G_0_/R_n_ fractions, the RMSE varied between 0.35 and 0.46 mm·d^−1^, which is within an acceptable range. This result is also in concordance with studies on SEB models performance for various crops showing that all tested models performed well with barley [11]. In comparison to the potato crop simulations, this concordance is mainly the result of a more accurate estimation of the evaporative fraction Λ by S-SEBI, reflecting the water consumption in accordance with the dynamics of the crop revealed by the NDVI range (Figure 5). This is confirmed by the R^2^ equal to 0.68 between Λ and NDVI for barley and the R^2^ value of only 0.28 for potato. The G_0_/R_n_ coefficient for models presenting the best performance when compared to HYDRUS are those leaving more available energy for the plant (G_0_ between 27 and 36 W·m^−2^) (scenarios S1B and S3B), in comparison to S2B (G_0_ = 113 W·m^−2^). Nevertheless, despite the high value for S2B, this did not have important effects on the ET_a_ calculation, as this model deviates from the two others after mid-April (Figure 6), when the plant starts to limit its water consumption due to the lack of precipitation. At the end of the season, the decrease of the evaporative fraction reflects the hydric stress of the rainfed barley detected by the biophysical variables of the vegetation cover (low NDVI in Figure 5).

## 4. Conclusions and Future Perspectives

This study aimed to assess the S-SEBI model’s performance based on remotely sensed data to determine ET_a_ for irrigated potato and rainfed barley in a semi-arid climate. First, we explored the ground-based calibrated HYDRUS-1D simulations of ET_a_. Results associated with model simulations indicated that the FAO-56 standard method is not recommended under water and salt stress conditions. Furthermore, seeding late potato achieved better ET_a_ (85%) than the early potato (67%). For the rainfed barley crop, it was noted that varying the LAI values at the final crop stage has a marginal effect on HYDRUS results, and the water stress effect is subject to water deficit intensity and its stage of development.

The S-SEBI model required mainly albedo and surface temperature within the experimental plot derived from Landsat 8 time-series images. The Λ based on empirical determination from each image, increased in the case of water availability reaching 0.80 for barley and 0.46 to 0.60 for potatoes at their maximum development, as shown by NDVI classes derived from Landsat 8 and Sentinel 2, respectively. The available energy (R_n_-G_0_) in combination with Λ allowed the computation of latent heat flux for the crops. The various scenarios of G_0_ calculation compared to the most used coefficient in the literature (G_0_/R_n_ = 0.2), revealed differences up to 117 W.m^−2^. Satellite-based modeled ET_a_ was compared to simulations by 1D calibrated HYDRUS. For both crops, the ET_a_ dynamic was correctly represented by S-SEBI, however values for potato were underestimated. The heterogeneity of Landsat 8 pixels mainly explains this as the potato is planted in rows spaced by soil. In the case of early and late potato, the least RMSE of ET_a_ was 1.5 mm·d^−1^, whereas for rainfed barley, the S-SEBI model was in perfect agreement with ground-based modeled ET_a_ with an RMSE not exceeding 0.46 mm·d^−1^. For both crops, the various scenarios of S-SEBI comparing G_0_/R_n_ methods showed a better performance of this ratio when it was in the range 0.2–0.3. Nevertheless, the good representation by S-SEBI for ET_a_ dynamics in both cases (irrigated potato and rainfed barley) and the fact that Λ reproduces water and salt stress is also a potential use for irrigation based on satellite images to differentiate the areas to be irrigated in priority. Despite its relative simplicity and the low data requirements, S-SEBI model results remain highly dependent on the empirical determination of dry and wet edges from the scatter plot pixels. Therefore, future studies need to address the exploration of the scatter space defined by (α,T_s_) in comparison to the spatial variation of vegetation cover and T_s_ [73]. Also, the comparison of energetic fluxes derived from HYDRUS instead of ET_a_ should be explored in comparison to S-SEBI model fluxes or to other SEB models, especially the dual-source models proven to make better estimation of ET_a_ in partially vegetative surfaces [8]. Besides, the outputs of such models could be used as inputs to the HYDRUS model since they afford evaporation and transpiration separately, thus providing a more realistic description of the main water and heat fluxes, as it was reported that drought and dry spells, characteristic of semi-arid climates, affect the transpiration rather than the evaporation considered constant [74].

Accurate assessment of spatiotemporal variations in soil moisture, evaporation, and transpiration is fundamental for water resource availability and sustainable management of scarce water resources in arid and semi-arid regions. Precise estimation of ET_a_ is still a challenging issue. However, the results in this study can be used to better plan and design precision agriculture for irrigated crops or deficit irrigation during the sensitive growth stages of rainfed crops.

## Figures and Tables

**Figure 1 sensors-23-02823-f001:**
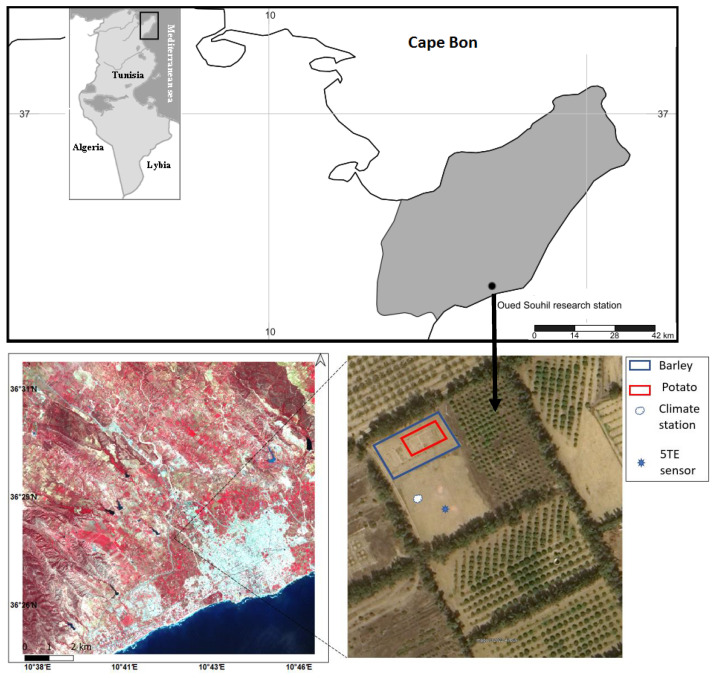
Geographic location of Oued Souhil experimental plot and surrounding area on a composite false color Sentinel 2 image (17 April 2017) (left) and Google Earth image (right).

**Figure 2 sensors-23-02823-f002:**
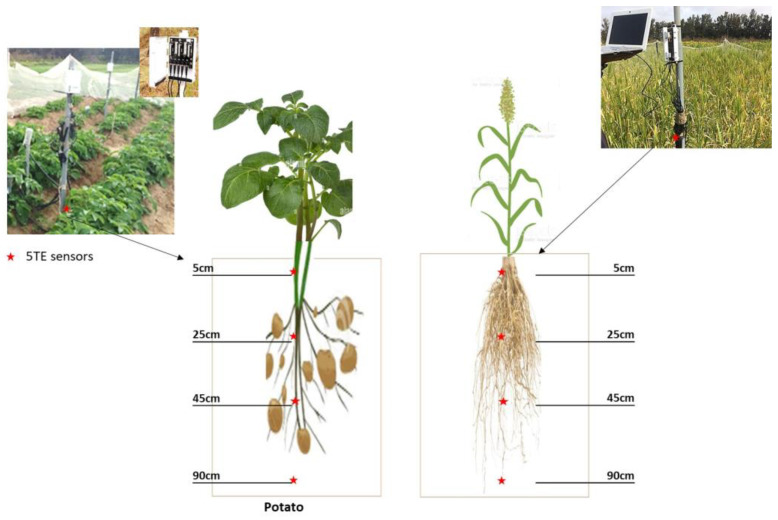
5TE sensor installation in potato and barley plots.

**Figure 3 sensors-23-02823-f003:**
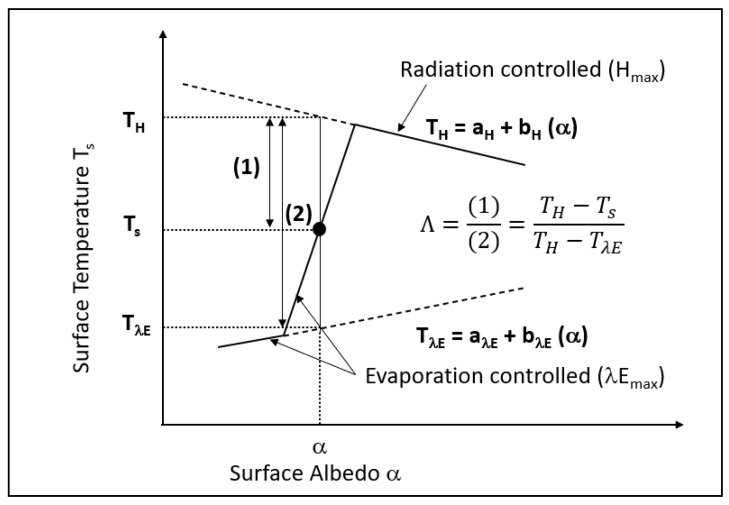
Schematic relationship between surface temperature and albedo in the S-SEBI algorithm (modified after [10]).

**Figure 4 sensors-23-02823-f004:**
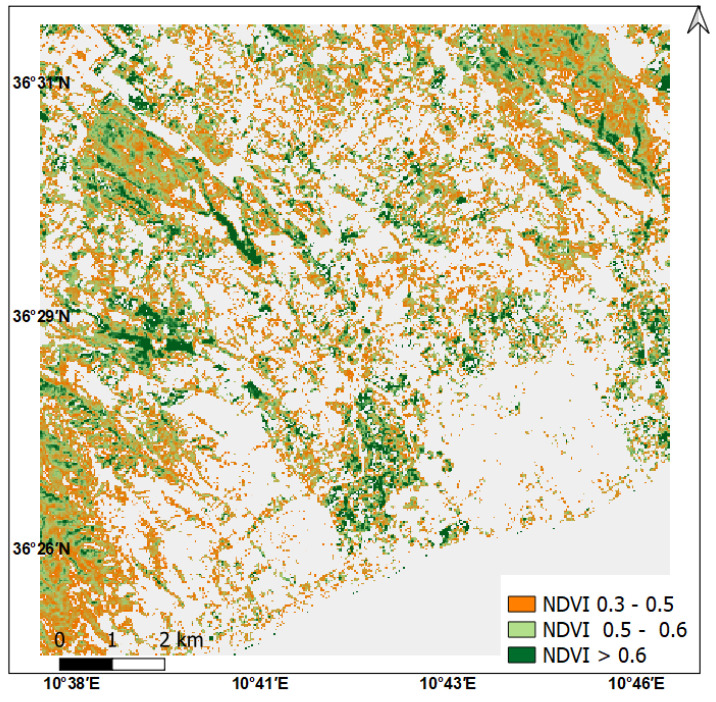
Example of an NDVI map and the corresponding scatter plot of albedo and surface temperature of sparse to full vegetation (NDVI > 0.3) around the experimental plot (NDVI and scatter from Landsat 8 image acquired on 28 February 2017).

**Figure 5 sensors-23-02823-f005:**
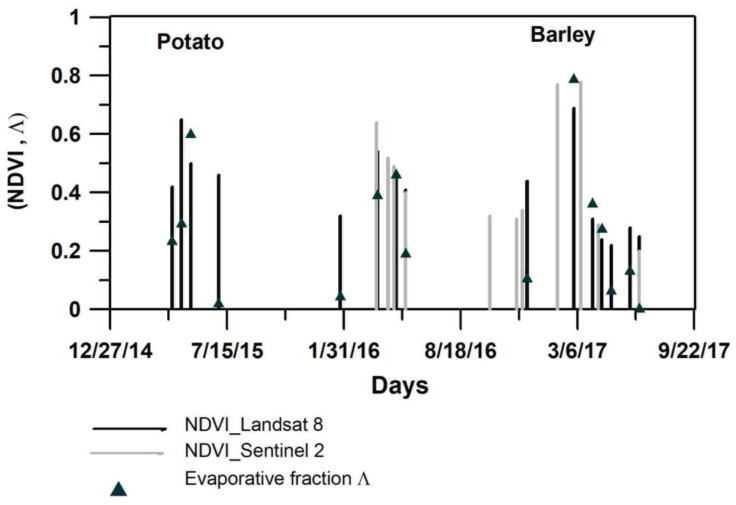
NDVI ranges from Landsat 8 and Sentinel 2 images and evaporative fraction range from S-SEBI in Oued Souhil plot.

**Figure 6 sensors-23-02823-f006:**
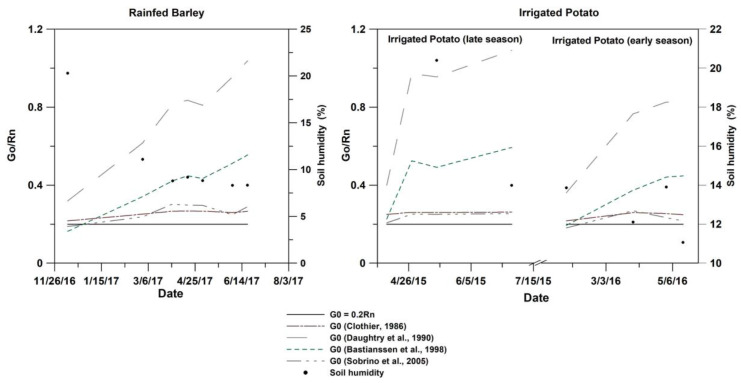
Variation of the ratio G_0_/R_n_ with different empirical formulas for irrigated potato (late and early growing seasons) and rainfed barley [36,37,38,39].

**Figure 7 sensors-23-02823-f007:**
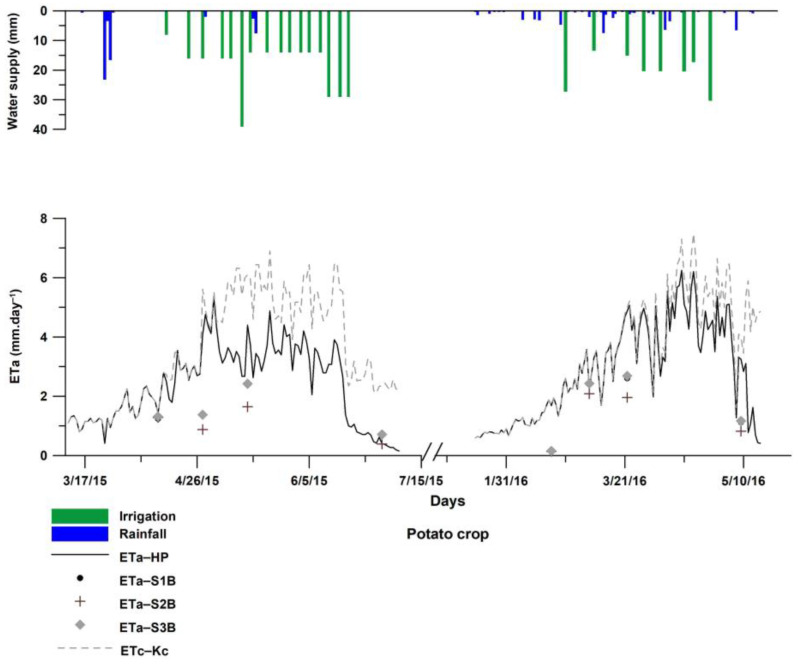
Comparison of ET_a_ by HYDRUS-1D, S-SEBI based on Landsat 8 and FAO-56 scenarios for early and late potato growing seasons.

**Figure 8 sensors-23-02823-f008:**
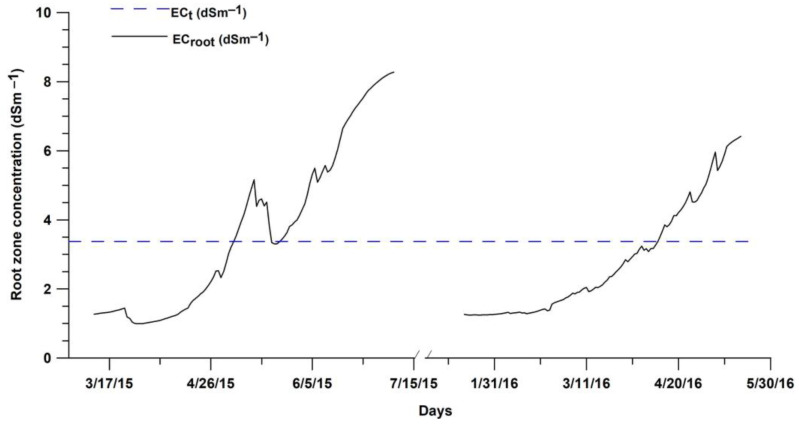
Root zone salinity variation for irrigated potato during two growing seasons for ET_a_–HP1 and ET_a_–HP2 scenarios.

**Figure 9 sensors-23-02823-f009:**
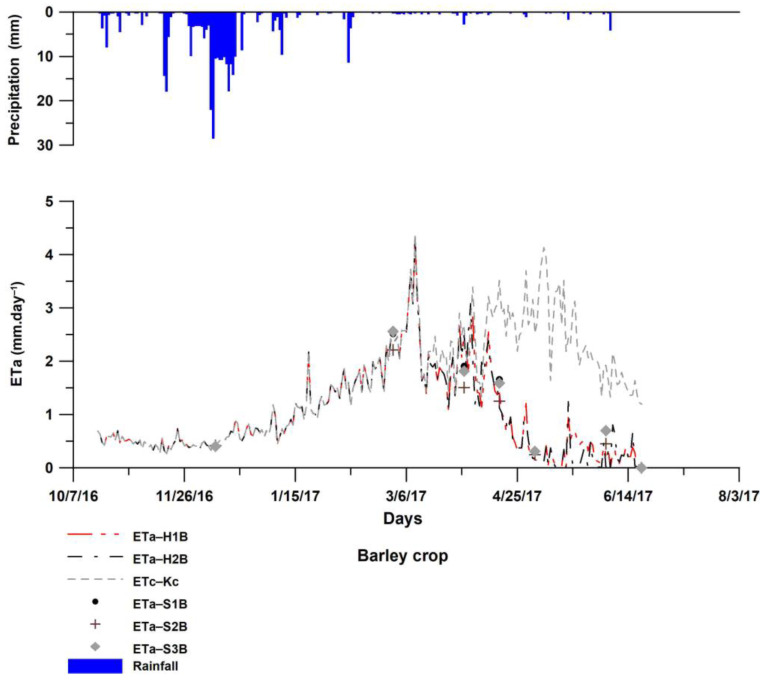
Comparison of ET_a_ by HYDRUS-1D, S-SEBI based on Landsat 8 and FAO-56 scenarios for rainfed barley.

**Table 1 sensors-23-02823-t001:** Particle size distribution of soil in the Oued Souhil experimental plot.

Soil Depth (m)	Clay (%) (d < 2 μm)	Silt (%) (2 ≤ d < 50 μm)	Sand (%) (50 ≤ μm d < 2 mm)	Bulk Density (g·cm^−3^)
0–0.2	4	25	70	1.41
0.2–0.4	15	11	73	1.52
0.4–0.6	16	12	71	1.69
0.6–0.8	19	11	70	1.73
0.8–1.0	17	11	70	1.81

**Table 2 sensors-23-02823-t002:** Irrigation, rainfall, and crop specific coefficient (K_c_) data.

	Season	1	2	3
	Crop	Potato: 11 March 2015 to 7 July 2015	Potato: 18 January 2016 to 17 May 2016	Barley: 18 October 2016 to 20 June 2017
Irrigation (mm)		336	170	0
Rainfall (mm)		20	70	291
ET_0_ (mm)		439	321	
K_c_	K_c-ini_	0.5	0.5	0.3
	K_c-mid_	1.15	1.15	1.15
	K_c-end_	0.75	0.75	0.25

**Table 3 sensors-23-02823-t003:** Calibrated soil hydrodynamic parameter.

Depth (m)	Θr (m^3^·m^−3^)	θs (m^3^·m^−3^)	α (m^−1^)	n (−)	Ks (m·day^−1^)
0–0.2	0.036	0.3938	3.32	1.69	2
0.2–0.4	0.0555	0.3947	4	1.60	1
0.4–0.6	0.0515	0.3571	3.14	1.50	0.28
0.6–0.8	0.051	0.3416	4	1.23	0.125
0.8–1.0	0.0507	0.3388	1	1.40	0.68

**Table 4 sensors-23-02823-t004:** Summary of parameter and ET_a_ estimation scenarios.

Model	ET_c_ Estimation	Earth Observation	Meteo	Soil/ Irrigation/	Scenario
S-SEBI	T_s_, ρ, α, NDVI Evaporative fraction Λ Soil fluxG_0_: Clothier (1986) [36] Daughtry et al. (1990) [37] Bastianssen et al. (1998) [38] Sobrino et al. (2005) [39] Sensible heat flux H Latent heat flux λE	Landsat 8 OLI TOA: Top of Atmosphere B2 [0.45–0.51 μm] (w_b_ = 0.293) B3 [0.53–0.59 μm] (w_b_ = 0.274) B4 [0.64–0.67 μm] (w_b_ = 0.233) B5 [0.85–0.88 μm] (w_b_ = 0.156) B6 [1.57–1.65 μm] (w_b_ = 0.033) B7 [2.11–2.29 μm] (w_b_ = 0.011) B10 [10.60–11.19 μm] B11 [11.50–12.51 μm] B2 to B7: 30 m res. B10 to B11: 100 m res. Sentinel 2 MSI TOC: Top of Canopy B4 [0.65–0.68μm] (10 m res.) B8 [0.78–0.90μm] (10 m res.)	Daily (T_a_, R_g_)		S1P1, S2P1, S3P1 S1P2, S2P2, S3P2 S1B, S2B, S3B
FAO-Kc	K_c_ Allan (1998) [6] ET_0_ Penman-Monteith, (1965) [23]		Daily (T_a_, R_g_, air humidity, wind speed)		FP1, FP2, FB
HYDRUS-1D	ET_0_ Penman-Monteith, (1965) [23] LAI potato Nasr et al. (2002) [48] LAI barley Afrasiabian et al. (2020) [49] LAI barley NDVI Sentinel 2 Boukari (2017) [50] Evaporation E Transpiration T		Precipitation	θ EC_p_ Irrigation doses EC_w_ Soil texture Soil hydraulic properties Crop parameters Feddes et al. (1974) [46]	HP1, HP2 H1B, H2B

**Table 5 sensors-23-02823-t005:** RMSE of G_0_ and λE of the various scenarios by S-SEBI for irrigated potato and rainfed barley, compared to reference case G_0_ = 0.2R_n_ (refer to Table 4 for scenarios).

RMSE [W·m^−2^] Scenario	G_0_–CLO S1	G_0_–BAS S2	G_0_–SOB S3	λE–CLO S1	λE–BAS S2	λE–SOB S3
Potato (8 dates)	25.09	117.51	19.42	9.29	41.99	7.62
Barley (7 dates)	27.36	113.70	36.75	7.72	25.31	9.34

## Data Availability

The data presented in this study are available on request from the corresponding authors. The data are not publicly available because they are currently used in ongoing theses.

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
