# Peer review of "Evapotranspiration in Semi-Arid Climate: Remote Sensing vs. Soil Water Simulation"

_sensors, 2023, doi:10.3390/s23052823_

Round 1

Reviewer 1 Report

This study assessed the performance of the S-SEBI model based remotely sensed data to determine ETa for irrigated potato and rainfed barley in semiarid climate using the calibrated HYDRUS-1D model. Results show that HYDRUS model is a fast and cost-effective assessment tool for water flow and salt movement in the crop root layer. ETa estimated by S-SEBI varied according to the available energy resulting from the difference between the net radiation and soil flux G0. The S-SEBI performed better for rainfed barley as compared to drip irrigated potato. This manuscript is well organized and in the scope of SENSORS. It is acceptable for publication after minor revision.

General comment

[1] The S-SEBI is used to estimate ETa for large spatial scale. In your study, Ts was at 100 m resolution. What resolution is for NDVI? The area of your experimental plot is only 500 m2. The different land use types around your experimental plot might impact the estimation of Ts using remote sensing. The difference in spatial scale between your experimental plot and the requirement of S-SEBI might results in the error for ETa estimation using S-SEBI. Please add some discussion in your R&D section.

[2] The precision of HYDRUS-1D model mainly depends on the hydraulic parameters, which are usually calibrated and validated using the measured soil water contents. Please add the hydraulic parameters for your soil type and provide the comparison of the measured and simulated soil water contents during the calibration and validation periods.

Specific comments

[1] ETa, not ETc, is used to represent actual crop evapotranspiration.

[2] In Table 2. There is large difference in irrigation water for potato in 2015 and 2016. Why? Please add the estimated ET0 for potato in 2015 and 2016 in Table 2.

[3] Some language improvement is necessary.

Author Response

Authors would like to thank the Reviewer 1  for the valuable comments. Please find answers in the uploaded file. 

Reviewer 2 Report

Thank you for this work, it seems a good applied development to help farmers with the decision making, but i dont see the novelty, i have been looking for both models and there are many papers where they are compared. I would like to know if this work has more differences among other than crop o country. Then, I will review the content.

Author Response

Authors would like to thank the Reviewer 2 . Please find answers in the uploaded file.

Reviewer 3 Report

The article is focused on the transfer of modern technologies to the process of determining selected soil parameters. The issue of evapotranspiration is not simple, while on the other hand, for selected crops, their irrigation regime affects them. In the article, the authors applied the surface energy balance model, which they compared with the transit model. Experimental measurements were made on research fields in a selected location in Tunisia. Two selected crops (potatoes and barley) were grown. Current soil moisture and pore electrical conductivity were monitored. The results point to the application possibilities of individual models. Based on the nature of the article, I would rather recommend another journal (Agriculture, Water, Applied Science, Remote Control) because the authors only used the model but did not propose any equipment and software.

Strengths side:

The article is at a enough level in terms of description of the issue, methodological procedures, processing of results and conclusions.

Author Response

Authors would like to thank the Reviewer 3 for the valuable comments. 

No specific comments mentioned by Reviewer 3
